# Music Therapy Enhances Episodic Memory in Alzheimer’s and Mixed Dementia: A Double-Blind Randomized Controlled Trial

**DOI:** 10.3390/healthcare11222912

**Published:** 2023-11-07

**Authors:** Shirlene Vianna Moreira, Francis Ricardo dos Reis Justi, Carlos Falcão de Azevedo Gomes, Marcos Moreira

**Affiliations:** 1Lato Sensu Postgraduate Program in Psychology, Faculty of Medical and Health Sciences of Juiz de Fora (SUPREMA), Juiz de Fora 36033-003, MG, Brazil; shirmoreira@icloud.com; 2Department of Psychology, Institute of Human Sciences, Federal University of Juiz de Fora, Juiz de Fora 36036-900, MG, Brazil; francis.justi@ufjf.br (F.R.d.R.J.); carlos.fagomes@gmail.com (C.F.d.A.G.); 3Department of Pharmacology, Institute of Biological Sciences, Federal University of Juiz de Fora, Juiz de Fora 36036-900, MG, Brazil

**Keywords:** music therapy, rehabilitation, dementia, memory, Alzheimer’s disease

## Abstract

*Objective*: This study aimed to assess whether a music therapy (MT) intervention could improve memory decline in older adults with and without cognitive impairment. A dual-retrieval model of episodic memory was employed to estimate memory processes. *Methods*: Forty-three older adults with a mean age of 76.49 years (*n* = 25 with Alzheimer’s disease (AD) and mixed dementia, and *n* = 18 healthy older adults) were randomly selected for the experimental and control groups. The study design was a double-blind randomized controlled clinical trial and a certified music therapist delivered the MT. The primary outcomes were measures of working memory, episodic memory, and autobiographical memory. *Results*: In the primary outcome measures, MT effects were restricted to episodic memory measures like the Figure Memory Test of the BCSB and the Speech and Sing Memory Test. In both tests, the experimental group improved from pre-test to post-test in delayed recall, but in the Speech and Sing Memory Test, the improvement was restricted to the AD and mixed dementia group. MT had no effects on the secondary outcome measures. *Conclusions*: These findings suggest that a structured MT intervention can be promising for rehabilitating episodic memory in older adults with dementia.

## 1. Introduction

Aging is a worldwide phenomenon and it is predicted that one out of six people will be aged 65 years and over in 2050 [1]. Although dementia is not part of normal aging and involves different risk factors (e.g., genetics, hypertension, cigarette smoking) [2], its prevalence in the older adult population is considered a public health problem [3]. Studies carried out in Brazil have shown a prevalence of around 12% in the population over 65 years of age. In Latin America, Alzheimer’s disease is considered the etiology of dementia in 49.9% to 84.5% of patients, followed by vascular dementia. In Brazil, the projected annual indirect costs were USD 13,468.80, 18,106.80, and 19,736.40 for mild, moderate, and severe forms of AD, respectively, representing 69 to 169% of family income [4]. Today, a third of older adults die of dementia and, in 2050, this disease will affect about 131.5 million people worldwide [3].

Studies with music therapy have shown efficacy in the treatment of behavioral and psychological symptoms of dementia, including agitation, irritability, depression, and apathy [5,6,7,8,9,10,11,12,13,14]. Recently, a series of systematic reviews and a meta-analysis converged on the positive effects of music therapy in reducing anxiety and depression [2,15,16,17]. In addition, practicing music (e.g., listening, singing, or playing an instrument) compensates for age-related declines in processing speed, memory, and cognition [18,19,20,21,22,23,24,25,26,27]. For example, Gómez Gallego and Gómez García (2017) found that after six weeks of music therapy, 42 patients with mild to moderate Alzheimer’s disease improved their scores on the MMSE, Neuropsychiatric Inventory, and Hospital Anxiety and Depression Scale. However, one shortcoming of the Gómez Gallego and Gómez García study was the lack of a control group in its pre-test and post-test design; thus, there was possible confounding between intervention and test repetition effects [20].

Han et al. also addressed the effects of music therapy on mental health [28]. This study employed a robust double-blind, randomized, placebo-controlled design and evaluated 64 older adult participants with MCI or mild dementia (CDR between 0.5 and 1) who underwent a broader multimodal cognitive enhancement intervention, which included music therapy. The intervention was effective in improving the measures of global cognition (MMSE and AD Assessment Scale—Cognitive Subscale) and promoted improvements in the frequency of problem behaviors in the Revised Memory and Behavior Problems Checklist. However, an important aspect to consider is that in this study, music therapy was only one part of a broader intervention; thus, the intervention results could not be unequivocally attributed to the music therapy part of the intervention.

Although recent systematic reviews and a meta-analysis presented mixed conclusions concerning the effects of music therapy on global cognitive functioning [2,15,16,17], another systematic review attempted to address the effects of music intervention on memory in Alzheimer’s patients [23]. In this systematic review, the authors considered only randomized controlled trials and found some positive effects of music intervention on memory. This may be related to music-related memory areas of the brain that, in the case of older adults with Alzheimer’s disease, are relatively spared [3,4,5,6,7,8,9,10,11,12,13,14,15,16,17,18,19,20,21,22,23,24,25,26,27,28,29,30]. Jacobsen et al. [30] carried out a study using functional MRI, demonstrating that the anterior caudal cingulate cortex and supplementary motor area were more activated when the participants heard known music excerpts. The same study was replicated with 20 patients with AD and 34 healthy controls without a history of neurological disease. The authors found that the pattern of degeneration in these cortical areas of people with AD was little affected and they had lower values of gray matter atrophy and hypometabolism in the rest of the brain. The deposition of ß-amyloid was not significantly lower in this area related to musical memory, suggesting this area was probably still at an early stage of the expected course of biomarker development [30,31,32]. These results may explain the surprising preservation of memory for music in AD.

Despite evidence favoring the effects of music interventions or music therapy interventions on memory [20,21,22,23,24,25,26,27,28], it is important to note that a good deal of these studies have only indirectly assessed memory, e.g., they have used MMSE scores to evaluate memory. This is an important shortcoming because the MMSE is more of a global screening test than a specific memory test. In addition, it is important to consider that memory is a broad category that can be applied to different memory systems (we could talk of working memory, semantic memory, episodic memory, autobiographical memory, etc.). Thus, it is important to have studies assessing these different memory systems to address whether music interventions have specific effects or not. Another issue consists of the specificity of the intervention itself, e.g., the study by Han et al. employed music therapy in the context of a broader cognitive intervention, while on the other hand, some studies did not employ MT but used other music-based interventions [33,34]. Considering that MT is a more specific intervention because it assumes a systematic use of various music-related methods delivered by a music therapist in the context of a therapeutic relationship, it is important to distinguish MT from other uses of music for health-promoting or recreational goals [35]. For example, at a very basic level, patients can passively listen to music as part of a recreational program. On the other hand, a music therapist engages the patient directly with music and musical instruments to achieve a specific clinical goal [36], e.g., aiming to stimulate working memory, a music therapist could ask a patient to reproduce certain sound sequences.

Considering the results of previous studies showing some effects of music therapy on memory [23] and data from studies showing that musical memory may be less impaired in AD [8,9,10,11,12,13,14,18,19,20,21,37,38,39,40], the present study aimed to assess whether music therapy could improve memory decline in older adults with or without cognitive impairment. This study provides advances in relation to previous studies by applying a strict music therapy intervention and assessing different memory systems with specific tests, including a theoretically oriented approach [41,42] to estimate memory processes in an episodic memory task. Therefore, if the intervention has any effect, we can better understand in which processes the effect is more relevant.

The present study was a randomized, double-blind clinical study that included a pre-test, MT intervention, and post-test. The MT intervention was conducted by the first author of the present study who is a certified music therapist and psychologist. The pre-test and post-test were performed by trained research assistants who were blinded to the group condition (experimental vs. control) and research hypotheses. The main goal was to investigate MT effects on episodic memory, autobiographic memory, and working memory. The secondary goal was to assess music therapy effects on executive function, mood, daily living activities, and caregiver burden. Although we expected a positive music therapy effect on memory due to the specific tailoring of our intervention, we considered the present study exploratory in nature because there is no clear empirical evidence or theoretical rationale about which memory processes should be affected by music therapy. The employment of different memory measures is one of the empirical contributions of the present work for better understanding music therapy effects on memory. Since we employed a specific theoretical model [41] to assess episodic memory data, we present below a brief characterization of this model.

### The Dual Retrieval Model of Episodic Memory

The theoretical model adopted in the present study was the dual-retrieval model [41,42,43,44,45], wherein there are two ways of recollecting episodic memories. The first is a vivid form where the retrieval of episodic memories is accompanied by the phenomenology of re-experiencing the target event, in which details of the context in which the event was experienced are mentally experienced. The second is a non-vivid form, which gives high confidence that events have been experienced but is not accompanied by a mental re-experiencing of the target event. The distinction between vivid recovery (recollective retrieval) and non-vivid recovery (non-recollective retrieval) has been explored over the past four decades to explain memory deficits in patients with brain injuries [25,26,27,28,29,30,31,32,33,34,35,36,37,38,39,40,41,42,43,44,45].

The theory of dual-retrieval by Brainerd et al. [46] uses Markov chains to measure the two recovery operations (direct access and reconstruction) and a familiarity operation, which is a slave to the reconstructive operation, strongly influenced by fuzzy trace theory [46,47]. This model, according to Gomes et al. [47], involves extracting simple quantitative measures from dual processes without the need for metacognitive judgments by analyzing recollection patterns over multiple tests.

Reconstruction is a recovery operation that uses essential traces of meaningful content to regenerate partially identified targets from the material studied [41]. The reconstruction is accompanied by a trial operation that performs confidence checks. The trial is a detection process that consists of a sign of familiarity [41]. When the construction provides a small set of candidate items, the items generate signs of familiarity, which are used to judge whether the item was present in the study phase according to the participant’s decision criteria. Although the familiarity judgment operation evaluates the products of reconstruction, this is a distinct process that can be affected by variables that do not affect the reconstruction. For example, increasing the items’ concreteness should hinder their reconstruction but increase the subjective familiarity of any reconstructed items [41].

Direct access is the quickest and most accurate of the two recovery methods and prevails at the beginning of a free recall protocol. In direct access, the reinstatement of the item is vivid and has realistic details. In general, it can be said that direct access represents the idea of recollective recall and that the processes of reconstruction and familiarity judgment represent the idea of non-recollective recall in different theories of dual processing [41,42,43,44,45,46,47,48].

In addition to modeling the processes involved in immediate study and test learning cycles, Brainerd et al. [38] point out that there are two independent forgetting operations. The first, called direct access (F_D_), consists of a process in which an item previously recalled via direct access can no longer be recalled. The second form of forgetting, called reconstruction (F_R_), consists of a process in which a previously reconstructed item can no longer be reconstructed based on partial information about it (for example, semantic information, like the meaning of the item). Considering this theory [38,41], we employed an episodic memory task specifically designed to assess familiarity, direct access, and reconstruction in immediate study–test learning cycles and forgetting.

## 2. Materials and Methods

### 2.1. Design

This was a randomized, double-blind clinical study that included a pre-test, music therapy intervention, and post-test. There were two groups of participants: the clinical group (with a diagnosis of the Alzheimer’s disease *continuum*) and the healthy group. Half of the participants of both groups were randomly allocated to the experimental group and the other half to the control group. The participants of both groups (experimental and control) did not have contact with each other and thus were blinded to the group condition. The pre-test and post-test were performed by research assistants who were blinded to the group condition (experimental vs. control) and research hypotheses.

### 2.2. Participants

Clinical group. Thirty older adults (over 60 years old) with a diagnosis of the AD *continuum* (mild cognitive impairment associated with AD, AD with vascular impairment (mixed dementia), or Alzheimer’s dementia) were recruited by convenience sampling at the Cognitive and Behavioral Neurology Unit (Hospital e Maternidade Therezinha de Jesus, Juiz de Fora, MG, Brazil) and at an Intermunicipal Cooperation Agency (ACISPES) in the same city. At the end of the study, a total of 25 older adults from the clinical group completed the pre-test and post-test. Those 25 older adults had a mean age of 79.08 (SD = 5.95) and 88% were female. The mean score on the CDR scale for this group was 1.34 (SD = 0.62) and the mean score on the MMSE was 14.68 (SD = 4.78). Considering the diagnosis, 17 out of 25 had a diagnosis of the AD *continuum* with a CDR between 1 and 2, but there were also five older adults with a diagnosis of amnesic MCI with a classification on the CDR scale equal to 0.5 and three older adults with mixed dementia (AD with vascular involvement). Each patient was assisted by a family caregiver.

The inclusion criteria for the clinical group were: (a) diagnosis of the AD *continuum* carried out by a specialist neurologist, with a CDR between 1 and 2 or a diagnosis of amnesic MCI with a classification on the CDR scale equal to 0.5; (b) being over 60 years old; (c) older adults who could be accompanied by a caregiver or family member during the sessions. The exclusion criteria were: (a) presence of cerebrovascular disease, including stroke; (b) severe AD; (c) traumatic brain injury; (d) uncorrected hearing or vision problems; (e) history of severe psychiatric illness; (f) drug and alcohol abuse; and (g) undergoing cognitive or psychotherapeutic rehabilitation.

Healthy group. The group of healthy older adults (*n* = 22) was recruited at the Senior Citizen Department of the city of Juiz de Fora, Minas Gerais, Brazil. At the end of the study, a total of 18 older adults from the healthy group completed the pre-test and post-test. Those 18 older adults had a mean age of 72.89 (SD = 7.36) and 94% were female. The mean score on the MMSE for this group was 24.72 (SD = 3.39).

Due to the limited number and specific nature of our population, i.e., people diagnosed with AD, we did not perform an a priori power analysis. However, the number of participants in the present study was quite close to the mean number of participants in other randomized controlled trials on the same topic [23].

### 2.3. Instruments

Scales and tests were used to assess the various types of memory as well as other cognitive and behavioral functions that are part of the symptoms of the older adult population with cognitive impairment (Table 1). These tests and scales were chosen because they were widely used in other music therapy intervention studies (e.g., MMSE, CDR, Geriatric Depression Scale, WAIS-III) [23] and/or because there is valid and reliable evidence for their use in Brazil [49]. The exceptions to this rule were the Musical Autobiographical Memory Test and the SASMET, which were constructed to assess autobiographical memory and episodic memory for verbal and musical materials in the present study.

#### 2.3.1. Tests for Participants’ Characterization

To characterize the sample, a sociodemographic evaluation form was used to collect data related to the gender, age, marital status, education, and socioeconomic level of the participants. Assessment of the CDR scale was performed by the attending physician. The MT evaluation protocol aimed to collect data on the sound-musical life of the participants. The MT protocol was applied as part of the first treatment session of the experimental group in this study.

#### 2.3.2. Primary Outcome Tests (Pre-Test and Post-Test)

To assess working memory, the Wechsler Adult Intelligence Scale III battery digits subtest was used. In addition to this, the Corsi block-tapping test was also used, which is an instrument like the digits subtest that assesses short-term memory, using the visuospatial modality. To assess episodic memory with visual material, the Figure Memory Test was used, which consists of 10 pictures from the BCSB and meets the requirements for validation with normalization and significant application in the Brazilian population [49].

To investigate episodic memory processes, the SASMET was developed in the present research to assess episodic declarative memory based on the dual-retrieval model of Brainerd et al. [38,41]. This test consists of 40 words with average frequency of occurrence. The words were balanced into two lists of 20 words each (List A and List B). These lists were counterbalanced in sung and spoken versions. Both lists (spoken and sung) were presented to the participant as a recording of the same duration (36 s), using 19 measures and a half, and a rhythmic speed of 62 beats per minute for the sung version. Valence and arousal [39], concreteness [40], the number of letters, the frequency of occurrence per million words, and the number of orthographic neighbors [50] were controlled so that the two lists did not have statistically significant differences in these dimensions (all *p* > 0.69). The sung and spoken versions were presented in two different sessions. Memory was assessed by playing the respective recorded word list (sung or spoken version) and asking the participant to recall the words, with the score being the correct number of recalled words. After five cycles of study–recall, a learning curve was constructed. Then, 20 min after the last recall attempt, the participant was asked to recall the words again. Distractors (other tests) were used during this 20 min interval to prevent any rehearsal attempts. Considering the dual-retrieval model of Brainerd et al. [32], based on the data from the study–recall cycles, individualized estimates of direct access (D), reconstruction (R), and familiarity judgment (J) were obtained for each participant. Since this test was developed for the present research, its reliability was assessed by correlating the control group’s scores on the SASMET in the pre- and post-tests. These test–retest reliability scores were 0.80 for the sung version and 0.92 for the spoken version.

The Autobiographical Memory Test [51] was used, in which the participants were asked to retrieve memories of their life histories from a series of 10 stimulus words, including five negatives and five positives. The Musical Autobiographical Memory Test was created by the first author through a pilot study. In this test, participants were instructed to remember specific events in their lives that were related to each of the 10 songs presented. The songs were presented in fragments that were between 25 and 30 s long. The report of memories for each music stimulus was recorded in audio. It was explained that a specific memory has temporal and spatial locations. The responses offered for each music stimulus were thus coded according to the three categories of the Autobiographical Memory Test [51]. Since the Musical Autobiographical Memory Test was developed for the present research, its reliability was assessed by correlating the control group’s scores on this test in the pre- and post-tests. This test–retest reliability score was 0.81.

#### 2.3.3. Secondary Outcome Tests (Pre-Test and Post-Test)

The MMSE was used to assess global cognition. It is a cognitive screening instrument that has criterion validity and reliability for the Brazilian version [52]. The Five Digits Test was used an instrument to assess the effect of attentional interference with conflicting information between number and quantity. It measures processing speed, attention, and executive functions (subcomponents inhibitory control and cognitive flexibility). The Shulman Clock Drawing Test was used for cognitive screening and measurement of dementia severity. It assesses planning, organization, constructive praxis, visuospatial skills, executive functions, and right hemisphere dysfunction with left neglect [53,54].

The Geriatric Depression Scale is an instrument validated for the Brazilian population and used to detect depressive symptoms in the geriatric population [55]. The Katz Functional Assessment Scale assesses basic ADL and has been cross-culturally adapted for use with the Brazilian population (Class II) and used in studies with people with dementia [49]. In addition, the Bayer Scale (Class II studies) was also used, which assesses instrumental ADL and has validation studies and diagnostic accuracy indices for AD [49]. The Informal Caregiver Burden Assessment Questionnaire has proven to be valid and reliable for measuring the physical, emotional, and social burden of caregivers of older adults and was adapted for Brazil [56].

### 2.4. Procedures

The procedures in this study adhered to ethical research policies and were approved by the Research Ethics Committee of the Federal University of Juiz de Fora, Brazil (CAAE: 83829418.0.0000.5147). For the clinical group, contact was made with the family caregivers inviting them to participate in this study, and all assessments and intervention activities were performed at home. For the healthy group, the participants were contacted directly at the Senior Citizen Department and the assessments and intervention activities were carried out in this same institution. Participants and/or family caregivers declared their consent to participate by signing an Informed Consent Form.

Pre- and post-test assessments were conducted by trained research assistants. The research assistants were undergraduate psychology students and blinded about which older adults were in the control group or the experimental group. In order to prevent the application of two musical tests in the same session (e.g., the sung version of the SASMET and the Musical Autobiographical Memory Test), the tests were applied in counterbalanced order according to four versions of counterbalanced assessment sheets, which were delivered in print to the research assistants according to the randomization. In both the pre-test and post-test, all instruments were applied in two sessions of about 40 min each. The post-test occurred about six weeks after the pre-test. The experimental group attended two weekly sessions of music therapy (with a maximum of one session per day), totaling 12 sessions of approximately 30 to 40 min each. The number of sessions in the present study was in line with the number of sessions in other studies that detected an effect of music therapy on cognitive functioning [16]. In addition, to ensure the fidelity of the strategies for treatment delivery, the same music therapist delivered all interventions and participants from the experimental group had to attend at least 75% of the music therapy sessions. All of the older adults included in the study were receiving conventional treatment with their respective attending physicians. Older adults with cognitive impairment were accompanied by caregivers. Caregivers received a handbook with relevant information about the disease, guidelines, and suggestions for cognitive stimulation activities to be performed at home, after the end of the sessions. The music therapy was conducted by the first author of the present study, who is a certified music therapist and psychologist. Below, we report the music therapy intervention according to the guidelines of Robb et al. (2011) [57].

#### 2.4.1. Music Therapy Intervention

A.Intervention Theory

The rationale for the music therapy intervention in the present study relied on two sources: (1) the evidence that, in the case of AD, music-related areas of the brain are relatively spared [3,29,30]; and (2) positive effects of music intervention on memory are achieved in people with Alzheimer’s disease [23]. Thus, in the present intervention, several music therapy techniques were employed to stimulate “musical encoding” of the information to be remembered (see Table 2 for a description of the techniques employed). Considering the impact of this stimulation on episodic memory processes, we did not expect an impact on reconstruction because this process depends mainly on semantic features and the music stimulation would focus encoding more on phonological and rhythmic aspects. However, considering that music-related areas of the brain are relatively spared, we expected the experimental group to profit from music during encoding to some extent. This could help in familiarity judgment (e.g., this word sounded like a studied word, or this word fit the melody) and to some extent decrease forgetting of well-encoded stimuli. It is important to note that our intervention strategy was compensatory and, considering disease progression, we did not expect memory improvements, only decreases in the decline rate.

B.Intervention Content

The music for each session was tailored based on patient assessment (see the music assessment protocol in the “instruments” section). Since the music was tailored to the patient’s musical preferences, there were a variety of songs employed in the intervention. However, most of these songs were popular Brazilian songs. The music was delivered both live and recorded. Live music was played by the music therapist and on some occasions by the older adults (see Table 1). For the recorded songs, playback equipment with speakers was employed. The speakers were placed on the participant’s side and the music therapist controlled the volume. Other intervention materials were figures, photos, calendars, daily listening to music, and sheets with musical activities. These activities were delivered to the older adults and caregivers to be carried out at home.

Table 2 presents a summary of the intervention strategies employed in the present study. It describes the activities and their objectives per session. Overall, these strategies were implemented through music listening, improvisation, and rhythmic auditory stimulation according to the objectives and activities described in Table 2. The intervention strategies respected the following sequence: (1) temporal and spatial orientation and verbal welcoming (introduction); (2) rehabilitation techniques in neurologic music therapy; (3) musical activity prescribed for the other days, and (4) assessment of the session by the music therapist and caregiver (closure). Figure 1 shows a photo from one of the music therapy sessions.

**Table 2 healthcare-11-02912-t002:** Music therapy intervention strategies used in the study.

Session	Theme	Activities	Objectives
1	Welcome	- Initial interview with patient and family/caregiver.	- Introduction of the therapist. - Therapeutic contract.
2	Introduction to patients	- Musical opening technique. - Calendar of sessions. - Seasonal orientation technique with music. - Technique of musical cues. - Practice technique distributed with music. - Caregiver-oriented musical activities.	- Temporal orientation. - Spatial orientation. - Episodic memory. - Autobiographical memory.
3	Identity	- Cover page. - Technique of musical cues along with the technique of learning without error. - Musical reminiscence therapy. - Musical closure technique. - Caregiver-oriented musical activities. - Music exploration technique.	- Temporal orientation. - Spatial orientation. - Episodic memory. - Autobiographical memory. - Visuospatial ability.
4	Musical memory	- Musical opening technique. - Seasonal orientation technique. - Music exploration technique. - Technique of musical cues along and technique of learning without error. - Musical reminiscence therapy. - Musical closure technique. - Caregiver-oriented musical activities.	- Temporal orientation. - Episodic memory. - Autobiographical memory.
5	Musical memory	- Musical opening technique. - Technique of musical cues. - Practice technique distributed with music. - Technique of attention to music with instruments. - Musical pictograms. - Musical closure technique. - Caregiver-oriented musical activities.	- Temporal orientation. - Episodic memory. - Attention. - Executive functions. - Visuospatial ability.
6	Musical memory	- Musical opening technique. - Musical reminiscence therapy. - Musical routine chart. - Musical closure technique. - Caregiver-oriented musical activities.	- Temporal orientation. - Episodic memory. - Autobiographical memory. - Working memory. - ADL.
7	Music in everyday life	- Musical opening technique. - Musical routine chart. - Memory game with music and images. - Musical closure technique. - Caregiver-oriented musical activities.	- Temporal orientation. - Episodic memory. - ADL. - Working memory.
8	Music in everyday life	- Musical opening technique. - Daily rhythm-oriented music technique. - Musical closure technique. - Caregiver-oriented musical activities.	- Temporal orientation. - Spatial orientation. - Attention. - Executive functions. - Working memory. - ADL.
9	Training memory and attention	- Musical opening technique. - Musical span. - Musical word memory technique. - Caregiver-oriented musical activities.	- Temporal orientation. - Working memory. - Attention. - Episodic memory.
10	Playing instruments	- Musical opening technique. - Use of musical instruments. - Musical closure technique. - Caregiver-oriented musical activities.	- Attention. - Humor. - Working memory.
11	Playing instruments	- Musical opening technique. - Music sequenced by numbers. - Musical closure technique. - Caregiver-oriented musical activities.	- Temporal orientation. - Attention. - Executive function. - Visuospatial ability.
12	Farewell	- Musical opening technique. - Musical recreation. - Musical closure technique. - Delivery of the caregiver’s handbook.	- Temporal orientation. - Episodic memory. - Working memory. - Caregiver activities.

ADL: Activities of daily living.

It is worth noting that, in this study, the sessions were designed according to what is presented in Table 2, considering the objectives outlined for the research, but with minor adjustments to the musical experience of each participant. Most sessions aimed at stimulating episodic memory (eight sessions), working memory (six sessions), and autobiographical memory (four sessions). In addition, temporal orientation (10 sessions) and attention (six sessions) were also stimulated. To a lesser extent, the following were also stimulated: spatial orientation (three sessions), executive functions (three sessions), visuospatial ability (three sessions), ADL (three sessions), and humor (one session). Additionally, caregivers received educational materials (one session).

The intervention took place in 12 sessions of approximately 30 to 40 min each. There were about two sessions per week, with a maximum of one per day. The intervention activities were performed at home for the clinical group and at the Senior Citizen Department for the healthy group. In both cases, the intervention took place in a private room and was individually delivered. For the clinical group, the intervention took place in the presence of the caregiver, who was instructed to maintain the musical activities at home. The caregiver was instructed and received a handbook entitled *Caregiver-oriented Musical Activities*.

#### 2.4.2. Data Analysis

Episodic memory processes were estimated from the SASMET scores based on the dual-retrieval model of Brainerd et al. [38]. The SASMET generates considerably more data for the participants because the study list consists of 20 words and there are five test–study cycles, in which participants hear the 20 words one at a time and then must remember them in free order. Thus, estimates of reliable individualized parameters can be determined using a sliding window bootstrap with a resampling procedure (T1T2T3 + T2T3T4 + T3T4T5) to produce more reliable estimates. The resulting estimates of D, R, and J are mean values of the separate parameters for each of these sequences [41]. This set of three parameters, D1, R1, and J1, measures direct access, reconstruction, and familiarity judgment in Trial 1, respectively, and a second set, D2, R2, and J2, measures the same processes in Trials 2 and 3 [58]. With these data, we could count the number of instances of each of the possible response sequences for each word (CCC, CCE, …, EEE), in which C indicates that the word was retrieved in the test and E indicates that the word was not retrieved. We could then enter these counts in the program, which returned parameter values and adjustment tests [58].

To analyze whether the dual-retrieval model fit the data of the present study, we used the same procedure as Gomes et al. [59], that is, testing the dual-retrieval model against a more complex model. This test produced a G^2^ statistic with one degree of freedom, with 3.84 as the critical value for rejecting the model adjustment in any experimental condition [59]. Thus, in Table 3, the values of G^2^ are presented for each experimental condition (group vs. diagnosis vs. time vs. modality) for the learning model and, in Table 4, we present the values of G^2^ for the forgetting model. Considering that the value of G^2^ in each experimental condition must be less than 3.84 for the model to be considered adequate for the data, we notice that the models of learning and forgetting passed the adjustment test, since the highest G^2^ value found was equal to 1.731. Thus, the individualized models were suitable both for healthy older adults and older adults with cognitive impairment, as well as for all groups (experimental and control), modalities (sung and spoken), and times (pre-test and post-test). This enabled direct comparisons of recovery processes between older adults with and without cognitive impairment.

## 3. Results

The mean age of the older adults was 76.49 years with a standard deviation of 7.18. The proportion of women was 91% with a standard deviation of 0.29. Regarding education, 7% of the sample was illiterate, 53.3% had one to four years of schooling, 18.6% had five to eight years of schooling, and 21% had over nine years of schooling. The Mann–Whitney U test was used to compare the age, years of schooling, and gender of the experimental and control groups. There were not statistically significant differences between the experimental and control groups for these variables (all *p* > 0.49); thus, these variables were not controlled in the following analysis.

### 3.1. Primary Outcomes (Effects of Music Therapy on Memory)

To evaluate the intervention effects, a 2 × 2 × 2 ANOVA was used with the time (pre-test vs. post-test), group (experimental vs. control), and diagnosis (healthy vs. clinical) factors as variables and, as a dependent variable, the score of the analyzed test in each case. The conventional criterion *α* = 0.05 was adopted in all reported statistical analyses. Because of the design we used, the effects of the treatment (experimental vs. control) were assessed by specific tests only when the time × group interaction (or any higher order interaction including these factors) was statistically significant.

A major diagnostic effect was observed on the scores of all primary outcome tests: Working Memory (Corsi block-tapping test), Autobiographical Memory (verbal- and music-based), and Episodic Memory (Figure Memory Test of the BCSB and SASMET). In all of those tests, as expected, the healthy group performed better than the clinical group. Because of the design we used, the effects of the treatment (experimental vs. control) were assessed by specific tests only when the time × group interaction (or any higher order interaction including these factors) was statistically significant. In none of the working memory or autobiographical memory tests was there a statistically significant time × group interaction or other higher order interaction, including time × group. This kind of statistically significant interaction was found only in the episodic memory tests; thus, in the following, we report only the results of those tests.

In the Figure Memory Test of the BCSB—Late Recall test, a statistically significant interaction between time and group was observed, *F* (1.39) = 5.005, *p* = 0.031, *µ_p_*^2^ = 0.114. Paired comparisons indicated that the experimental group improved from the pre-test (*M* = 5.460) to the post-test (*M* = 6.364), *F* (1.39) = 8.527, *p* = 0.006, *µ_p_*^2^ = 0.179, which did not occur in the control group (*p* > 0.86). This interaction can be seen in Figure 2.

In the memory processes estimated by the SASMET task, considering direct access, we did not observe a time X group interaction or any higher order interaction including these factors). In familiarity judgment, there was a statistically significant interaction between modality, time, and group, *F* (1.39) = 5.090, *p* = 0.030, *µ_p_*^2^ = 0.115. Paired comparisons between the control and experimental groups showed that, in the sung modality, the control and experimental groups did not differ in the pre-test, *F* (1.39) = 0.029, *p* = 0.865, *µ_p_*^2^ = 0.001 (*M* = 0.459 and *M* = 0.442, respectively). However, in the post-test, the experimental group performed better, *F* (1.39) = 4.77, *p* = 0.035, *µ_p_*^2^ = 0.109 (*M* = 0.338 and *M* = 0.508, for the control and experimental groups, respectively), which did not occur in the spoken modality. In reconstruction, we also noticed a major effect of modality, *F* (1.39) = 38.533, *MSE* = 0.017, *p* < 0.001, *µ_p_*^2^ = 0.497, and reconstruction was greater in the spoken modality (*M* = 0.217) than in the sung modality (*M* = 0.127).

When assessing forgetting in the SASMET, we noticed a major effect of forgetting type, *F* (1.39) = 7.985, *MSE* = 0.141, *p* = 0.007, *µ_p_*^2^ = 0.170, wherein people forgot more in direct access forgetting (*M* = 0.406) than in reconstruction forgetting (*M* = 0.081). There was also a statistically significant interaction between diagnosis, group, and time, *F* (1.39) = 6.417, *p* = 0.015, *µ_p_*^2^ = 0.141. Paired comparisons in the clinical experimental group between the pre-test (*M* = 0.332) and post-test (*M* = 0.160) indicated that forgetting decreased in this group, *F* (1.39) = 10.802, *p* = 0.002, *µ_p_*^2^ = 0.217. In the other groups, there was no statistically significant difference between the pre-test and post-test (all *p* > 0.13). This interaction can be seen in Figure 3.

### 3.2. Secondary Outcomes

Most of the assessed older adults, especially in the clinical group, had difficulties understanding and performing the Five Digits Test. Thus, due to lack of data and poor performance, it was not possible to analyze the data from this test.

Caregivers in the clinical group answered the questionnaires on ADL, including the Katz Functional Assessment Scale, Bayer Scale, and Informal Caregiver Burden Assessment Questionnaire. To analyze the scores of the ADL scales, the non-parametric Mann–Whitney test was performed for two independent samples, comparing the control and experimental groups, both in the pre-test and post-test. There were no differences between the groups in the pre- and post-tests.

Considering the MMSE, Shulman Clock Drawing Test, and Geriatric Depression Scale, a 2 × 2 × 2 ANOVA was used with the time (pre-test vs. post-test), group (experimental vs. control), and diagnosis (healthy vs. clinical) factors as variables and, as a dependent variable, the score of the analyzed test in each case. The conventional criterion *α* = 0.05 was adopted in all reported statistical analyses.

A major diagnostic effect was observed on the MMSE scores and Shulman Clock Drawing Test scores, with healthy older adults performing better than those in the clinical group. There was not a diagnostic effect on the Geriatric Depression Scale scores. There was not a statistically significant time × group interaction or other higher order interaction, including time × group, in any of the above tests (MMSE, Shulman Clock Drawing Test, and Geriatric Depression Scale). Thus, we could conclude that the intervention had no effects on these tests.

## 4. Discussion

As expected, the healthy older adults’ group had better performance than the clinical group in almost all variables, except for GDS scores. In the Figure Memory Test of the BCSB—Late Recall, which is an episodic memory test, the experimental group improved from pre-test to post-test, while the control group did not show any improvement. In addition, considering the episodic memory processes estimated by the SASMET task, in familiarity judgment in the sung modality, the experimental group performed better than the control group in the post-test. More importantly, forgetting decreased among older adults from the experimental clinical group compared to older adults from the control clinical group. It is interesting to note the converging results from delayed recall tasks like the Figure Memory Test of the BCSB—Late Recall (about five-minute interval between last study and recall) and forgetting measures in the SASMET task (about 25 min interval between last study and recall), in which the intervention improved episodic memory. Thus, the MT intervention effects were very specific and affected only episodic memory measures. In the other variables included in the study, no effects of the MT intervention were observed, that is, the interaction between time vs. group or a higher order interaction including these factors was not statistically significant. We can then conclude that the intervention had no effects on the following tests: screening test of the MMSE, Katz Functional Assessment Scale in ADL, Geriatric Depression Scale, Wechsler Adult Intelligence Scale III battery digits subtest, Corsi block-tapping test, Autobiographical Memory Test, Musical Autobiographical Memory Test, Five Digits Test, Shulman Clock Drawing Test, and Informal Caregiver Burden Assessment Questionnaire.

In the Figure Memory Test of the BCSB in the recall condition (after an interval of five minutes), there was a significant interaction between time and group. This showed that the MT intervention had some effect on recall. On the other hand, the intervention did not seem to affect tests of incidental memory or figure recognition memory. It is necessary to consider that, in the literature [60,61], late recall is a good predictor of dementia, with sensitivity and specificity greater than 80%. Thus, the effect of the intervention seemed to have been detected precisely in the measure with better sensitivity and specificity. Nitrini et al. [61] suggested that, when the items to be retrieved are presented as simple drawings, coding is easier for illiterate individuals and those with low education levels. This may explain the reasonable performance of older adults participating in the present study in the Figure Memory Test of the BCSB in the recognition phase, since the sample consisted of 60% older adults with low education levels.

The SASMET was used because of the specificity of the present experiment, which sought to assess episodic memory using the dual-retrieval theory. Unlike direct access, in which the intervention did not seem to have had any effect, the statistically significant interaction between modality, time, and group in familiarity judgment indicated the possible effect of music therapy intervention. In this case, when the control group was compared with the experimental group in the sung modality, we observed that the experimental group performed better in the post-test. This result is probably a protective effect of the intervention. Another interesting aspect is the fact that this result was restricted to the sung modality, considering the music therapy intervention. In this sense, it is possible that the intervention may have led the participants, even if incidentally, to encode the stimuli in the sung versions in terms of the sounds (e.g., the rhymes), associating them with the melody and rhythm of the music. In reconstruction (R), which is the probability that an item that was not retrieved by direct access will be reconstructed, there was a major effect of modality, and reconstruction was greater in the spoken modality than in the sung modality. This result was expected because reconstruction draws on semantic features to generate candidate stimuli and, since the intervention emphasized the phonological aspects of the stimuli, we did not expect any intervention effects on reconstruction.

It should be noted that, in the figure memory task, there was also no effect of the intervention on immediate recall (learning) tests, only on late recall tests (after five minutes). Thus, it is important to consider the data from the forgetting model of the SASMET task. In the dual-retrieval model, forgetting was assessed considering the last stage of the task of sung and spoken words, which considered the number of words retained in a late recall (approximately 25 min after the last essay) and without opportunity of a new trial. There was a major effect of the type of forgetting, with older adults forgetting more in direct access (F_D_) than in reconstruction (F_R_). This was already expected since direct access is more easily lost over time [38]. An interesting finding was the observation of a significant modality effect in the clinical group, that is, they forgot less in the sung version in the post-test in relation to the pre-test, regardless of the type of forgetting. Combined with the statistically significant interaction between diagnosis, time, and group, in which forgetting in the experimental clinical group decreased in the post-test in relation to the pre-test, this fact indicates the possible positive effects of the music therapy intervention on the clinical group. This result is potentially interesting as it aligns with the hypothesis that areas related to musical memory are less impaired in AD [30,31,32,33,34,35]. For the clinical group, since episodic memory was deficient, information encoded based on sounds may have remained a useful resource for a longer time and helped with late recall (forgetting) due to the lack of available semantic information. This is in accordance with the MT intervention activities, as they focused encoding more on phonological and rhythmic aspects. For example, this could help in familiarity judgment because the participants could rely on information about whether a word sounded like a studied word or whether the word fit the melody. To some extent, this also could decrease forgetting because participants could rely on sounds as an additional source of encoding information, and this would help in recall, especially in the absence of semantic information. This interpretation and the present results are in line with the results observed by Palisson et al. [62], who sought to investigate the encoding and retrieval of sung texts compared with spoken texts. The authors observed that sung texts were better remembered than spoken texts among individuals with AD. Thus, the present study extends previous findings by showing that an MT intervention can improve episodic memory in people with dementia by stimulating “musical encoding” of new information.

It is important to note that our intervention strategy was compensatory; we tried to stimulate “musical encoding” because of the evidence that areas related to musical memory are less impaired in AD [30,31,32,33,34,35]. In this sense, one of the main results reported by Basaglia-Pappas et al. [63] was that memory for melodies seemed to be spared in mild AD. It is important to consider that, in the study by Basaglia-Pappas et al. [63], not only recovery but also melodic recognition of participants with Alzheimer’s disease did not show any significant difference from participants without AD, which indicates the preservation of this type of information in their memory.

Considering working memory and autobiographical memory, which were also assessed in the present study, there were no effects of the intervention on these memory systems. One possibility for the lack of intervention effects on working memory measures like the Wechsler Adult Intelligence Scale III battery digits subtest and the Corsi block-tapping test is that working memory training has very specific effects that do not transfer to other tasks (for a meta-analysis, see Melby-Lervåg et al. [64]). Since our participants were not specifically trained on the Wechsler Adult Intelligence Scale III battery digits subtest and the Corsi block-tapping test, they may have failed to transfer the trained strategies to these tasks. Considering the autobiographical memory tasks (Autobiographical Memory Test and Musical Autobiographical Memory Test) it is important to note that on those tasks, the word or song stimulus was supposed to be used as a clue for recalling a specific autobiographical memory encoded a long time ago. This is different from the processes used on the Figure Memory Test of the BCSB and the SASMET test. On these tests, the participants had a study cycle in which they could employ “musical” strategies to encode the target items. Thus, considering the hypothesis that participants’ performance improved because they used “musical encoding” as a compensatory strategy when they were studying the target items, they would not be able to do this in the case of the Autobiographical Memory Test because the items were only clues to recall episodic memories encoded long ago.

Finally, it important to note that the MT intervention did not have any effects on secondary outcome measures like global cognition (MMSE), executive functioning (Shulman Clock Drawing Test), mood (Geriatric Depression Scale), daily living activities (Katz Functional Assessment Scale, Bayer Scale) or caregiver burden (Informal Caregiver Burden Assessment Questionnaire). In this sense, it is important to consider that the MT intervention in the present study was basically focused on memory stimulation (in its different systems and processes), with few sessions to stimulate other aspects such as, for example, humor. Therefore, any improvements in these other aspects should be due to an improvement in memory, that is, it was expected that if there was a significant improvement in memory, activities that depend on memory, such as ADL, would also improve. The point is that, as in AD, memory deficit is progressive [65,66], and the most that could be expected was a decrease in the rate of memory loss, that is, less worsening in the experimental group compared to the control group. Therefore, it is understandable that, in the present study, no improvements were detected in aspects such as mood, daily living activities, or caregiver burden.

One of the limitations of the present study was not having a more homogeneous group of participants. Like in the majority of studies [16,23], the clinical group was relatively heterogeneous and formed by older adults with MCI associated with AD, mixed dementia (AD with vascular involvement), and Alzheimer’s dementia. Although these clinical forms may form part of the Alzheimer’s disease *continuum*, we cannot be fully certain as to whether the results found apply identically to all of these subgroups. Thus, it would be important to carry out future studies with a more homogeneous sample of people with AD. Another issue regarding the participants in the present study is that about 90% were women. Although it is common to have more women than men diagnosed with AD, this imbalance hinders the generalizability of our results. In addition, it would be important to have some follow-up after a significant period to better monitor whether any gains in memory could also impact other cognitive and mood aspects. Finally, as stated by Shadish et al. [67], it is important to consider that finding an effect in an intervention study, however well-planned it may be, only indicates that the full package of the intervention (the established rapport, researchers’ personal skills, specific techniques used, receptiveness of the participants, among others) worked. Thus, when the efficacy of an intervention is detected, it is always important that future studies better explore which specific aspects of the intervention are crucial for success of the treatment.

## 5. Conclusions

In the present study, the music therapy intervention had a positive effect on reducing forgetting in episodic memory tasks. The results of the present study suggest that memory for sound/musical aspects of stimuli may be better preserved than memory for other aspects in people with AD, converging with data from other studies [30,31,32,33,34,35]. Thus, music-based treatments can be considered promising in the stimulation of episodic memory in older adults with dementia.

## Figures and Tables

**Figure 1 healthcare-11-02912-f001:**
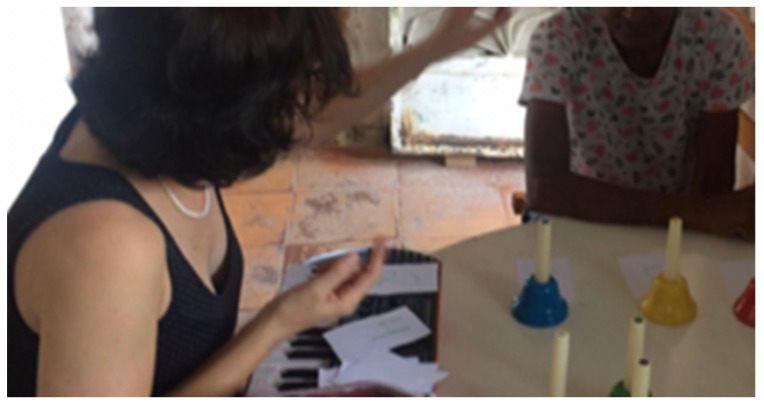
Photo from one of the music therapy sessions. In this session, the patient had to identify and repeat some sound sequences using instruments.

**Figure 2 healthcare-11-02912-f002:**
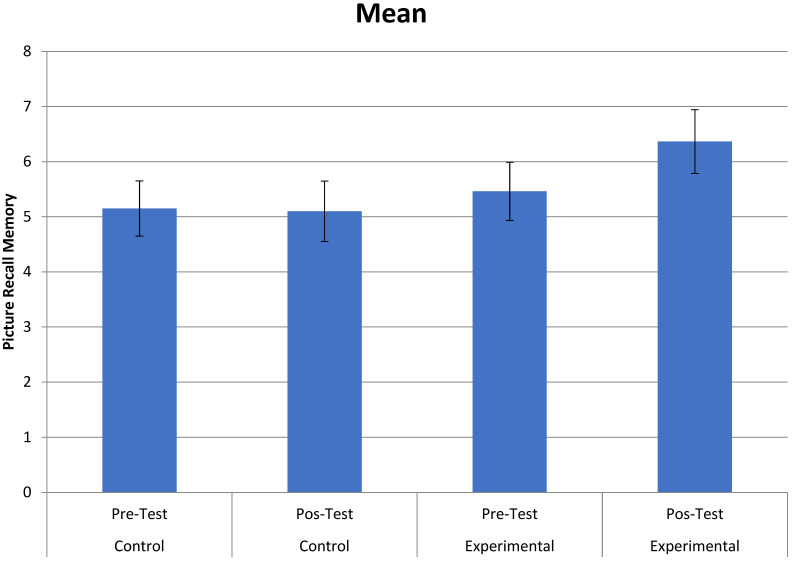
Interaction between group (control and experimental) and time (pre- and post-tests) in the Figure Memory Test of the BCSB—Late Recall.

**Figure 3 healthcare-11-02912-f003:**
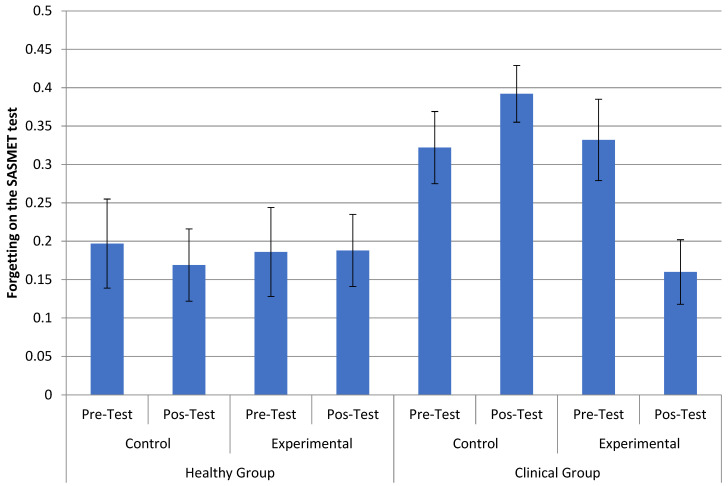
Interaction between diagnosis (healthy and not healthy), group (control and experimental), and time (pre- and post-test) for the cycle of direct access forgetting (F_D_) and reconstruction forgetting (F_R_) in the tasks of sung and spoken words.

**Table 1 healthcare-11-02912-t001:** Scales and tests that assessed cognitive and behavioral functions in the study.

Assessment	Tests
Questionnaire	Music-based therapy protocol (applied on the first session of music therapy, only in the experimental group)
Rating	CDR (provided by the attending physician)
Global Cognitive Assessment	MMSE
Functional	Katz Functional Assessment Scale (applied on family caregivers, only in the clinical group)
	Bayer Scale (applied on family caregivers, only in the clinical group)
Humor	Geriatric Depression Scale
Specific cognitive areas	*Memory*
	Figure Memory Test of the BCSB
	Autobiographical Memory Test
	Musical Autobiographical Memory Test
	SASMET
	*Attention and Executive Function*
	Five Digits Test
	Corsi block-tapping test
	Wechsler Adult Intelligence Scale III battery digits subtest
	*Visual Perception*
	Shulman Clock Drawing Test

**Table 3 healthcare-11-02912-t003:** Adjustment of the dual-retrieval model for each experimental condition (group vs. diagnosis vs. time vs. modality) of the learning model.

			Sung Words	Spoken Words
Group	Diagnosis	Time	Mean G^2^	Mean G^2^
Control	Healthy	1	0.690	0.317
		2	0.989	1.452
	not healthy	1	0.381	0.740
		2	0.400	0.680
Experimental	Healthy	1	0.643	1.079
		2	0.473	0.512
	not healthy	1	0.107	0.433
		2	0.668	1.406

**Table 4 healthcare-11-02912-t004:** Adjustment of the dual-retrieval model for each experimental condition (group vs. diagnosis vs. time. vs. modality) of the forgetting model.

			Sung Words	Spoken Words
Group	Diagnosis	Time	Mean G^2^	Mean G^2^
Control	Healthy	1	0.734	1.147
		2	1.376	1.425
	not healthy	1	0.358	0.435
		2	0.478	0.122
Experimental	Healthy	1	1.235	1.731
		2	0.948	1.556
	not healthy	1	0.686	1.210
		2	0.706	0.396

## Data Availability

The data that support the findings of this study are available upon request from the corresponding author. The data are not publicly available due to information that could compromise the privacy of research participants.

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
