# Peer review of "Music Therapy Enhances Episodic Memory in Alzheimer’s and Mixed Dementia: A Double-Blind Randomized Controlled Trial"

_healthcare, 2023, doi:10.3390/healthcare11222912_

Round 1
Reviewer 1 Report
Comments and Suggestions for Authors
In the Introduction section, could you please briefly give examples of other than music intervention in this field?
In the Methods section, could you please explain the rational behind choosing the specific assessment instruments?
It would be interesting to see a photo from a music therapy session if possible.
It would be helpful to have a mention on the difference between using Brazilian music and other type of music in music therapy sessions.
Author Response
Dear reviewer,
Thank you for your encouraging review. In the manuscript's revised version, we incorporated all your suggestions. We included:
- Examples of intervention strategies.
- We explained the rationale for the assessing instruments.
- We included a photo from a music therapy session.
- Considering the issue of Brazilian music, It is important to note that the music therapy intervention was tailored to the participant's music history. Using familiar songs was a way to engage the participant in the music therapy sessions; however, the type of music used did not change the activities they had to do.
Sincerely,
the authors
Reviewer 2 Report
Comments and Suggestions for Authors
Dear Authors
The manuscript, although it presents a low number of participants, offers a study of a strategy such as Music Therapy (MT) that could influence the maintenance or improvement of cognitive processes in older adults, especially episodic memory, very relevant in the consideration of possible deteriorations. that can evolve to more severe pathologies, especially Dementia or Alzheimer's Disease (AD). The manuscript is too extensive for the findings that are reported, which is why it is desirable to summarize the fundamental results (limiting ourselves to the finding that was partially achieved) and in the discussion (analyzing the finding and how it contributes to scientific knowledge on the subject, to the light of the available literature regarding the topic investigated. This is in general.
Specifically, it presents some observations to consider that could clarify some aspects exposed in the manuscript.
The introduction provides information regarding the study variables, in this case, the influence of MT in older adults diagnosed with dementia. The writing of this section establishes the gaps regarding studies where TM specifically, and not just incorporations of music in broader therapies, can influence an intervention action in people with cognitive disorders. It also interestingly exposes the changes in brain regions that are reported and that, in some way, show the influence of this variable on the nervous system until now. At a neuropsychological level, the contributions related to the classes and tests used are relevant because they are validated to screen basic cognitive functions, in addition to highlighting the lack of this type of evidence in the scientific literature. In any case, the instruments chosen and used to evaluate cognition linked to the intervening variable such as MT should be more fully justified.
Likewise, explaining more precisely what the characteristics of an action are through the musical resource that can be called “Music Therapy”, would be important to specify it to distinguish it from other procedures, especially the one shown in this manuscript, which is presented later in a table. In this, expand and justify what was mentioned between lines 88 to 92.
Some specific observations to the introduction:
Line 33-41. Although they refer to Van der Steen, more focused on the fact that dementia in older adults is a global mental health problem, I suggest NOT attributing that aging brings with it cognitive deterioration. Precisely at present, the evidence shows an ageism bias in relating the elderly to said deterioration. The phenomenon exists, but there is no conclusive cause. See among others:
doi: 10.1093/geront/gnw194;
DOI: 10.1159/000508660;
DOI: 10.3390/ijerph18083988
Line 42-51: Reports evidence of the effects of music therapy in people diagnosed with AD, and also highlights the lack of explanatory studies with a control group. However, there is a history of previously executed work that should be updated. For example, among others from different magazines:
doi: 10.1016/j.nrl.2014.11.001;
doi: 10.3390/geriatrics5040062;
doi: 10.3389/fmed.2020.00160
In materials and methods, some observations:
The instruments reported to evaluate cognitive aspects and determine the neuro normative diagnosis such as dementia are adequate. However, it is suggested to include the descriptive statistics obtained from each group in order to have references to the cognitive state of the participants.
Line 172-173. Three diagnoses are mentioned for the clinical or experimental group, with a predominance of women (line 179) and then in line 180 an inclusion criterion is mentioned. This clinical or experimental group should be described in very precise terms and only once, to avoid confusion. The question remains whether the other diagnoses were also considered. In fact, in the results and discussion, there is very little talk of mild cognitive impairment.
Line 176: What diagnostic condition did the older adults considered in the experimental group have?
Line 177: It says that the participants completed 75% of the treatment sessions. It is advisable to reference the criterion or bibliographic basis that supports that said 75% was sufficient to impact the results.
The number of scales and tests for adults with cognitive impairment is striking. What was the order of administration, is it according to Table 1? Could fatigue or cognitive load have played a role at some point? How did you control these aspects? This is regardless of the approval of the Ethics Committee in line 267, which approves the procedure. It is suggested to incorporate the administration procedure of each test into a timeline, both in the pre-TM and post-TM conditions. This would make it easier to understand the clarity and relevance of the procedure.
Line 330-335: The stimulated neuropsychological functions mentioned are 10. What relationship did the WM procedures have with the tests used to evaluate these functions, since Table 1 presents several, more than ten tests or scales?
For example, on Line 337: One per day, or did it have another distribution? In relation to the test administration times, what was the delay in this regard?
In the Results and Discussion point.
The graphs should have the error bars that can be obtained from the application of the 3-way ANOVA in a statistical program.
What background allows us to discuss this result and this interpretation? Is it possible to compare it with evidence obtained in other studies? This could inform the discussion of the findings obtained. For example, schooling and the relational level are important in improving cognitive performance given the stimulation of the systems that favor neuronal neuroplasticity, based on the cognitive reserve that adults in this age range could present. The question that arises is whether verbal or non-verbal coding weighs more.
In music, it seems that non-verbal influences. Obviously, the interaction between the two favors the evolution of cognitive processes. Another question that remains, is whether there is evidence in the literature that in the 5 tests mentioned there are no significant differences compared to a powerful stimulation such as MT.
Although the contents of the tests are not directly related to WM, could the training of cognitive processes through WM strategies favor them?
In this regard, could TM be intended for the training of these tests? This, clearly considering the conservation of musical coding regions in people with AD. On the other hand, was their life-historical relationship with music considered in the previous study of the participants? And with what kind of music? Or is it just some aspects such as rhythms, tones, and harmonies, among other aspects of music?
The coding aspect involves verbal or non-verbal concepts. This requires prior learning processes, which is why association aspects must be key. The question is whether this TM favors those types of associations that allow it to respond or maintain functionality, which is key in terms of episodic memory.
Line 615-621: Was it considered a way to evaluate the role of the caregiver, mediator, or family member, in terms of their TM intersession intervention? If so, the role of these people in TM could be considered. There is evidence that they are fundamental and that they influence. The question is, according to what UDS mentioned, what aspects of episodic memory could its role in WM be focused on that allow the results obtained in people with AD?
Finally, it is suggested that the conclusion be oriented towards the findings of this research, mentioned punctually and clearly. How does WM contribute to episodic memory, as observed? What aspects of TM are key in this process? How could the role of TM as a preventive strategy for cognitive decline be improved?
Author Response
Dear reviewer,
Thank you for your encouraging review. Your comments and suggestions were addressed in the manuscript's revised version. We believe the manuscript's revised version represents a significant improvement from the previous one. Below, we present your comments and how we addressed them.
Reviewer: "In any case, the instruments chosen and used to evaluate cognition linked to the intervening variable such as MT should be more fully justified."
- Answer: In the instruments section, we provided the rationale for the assessing instruments and references concerning their reliability and validity evidence. In addition, we included test-retest reliability for the instruments we developed.
Reviewer: "... explaining more precisely what the characteristics of an action are through the musical resource that can be called “MusicTherapy” ... In this, expand and justify what was mentioned between lines 88 to 92.".
- Answer: Considering the cited passage (lines 88 to 92), we included concrete examples to help differentiate the general use of music from the use of music in music therapy.
Reviewer: "lines 33-41 ... I suggest NOT attributing that aging brings with it cognitive deterioration."
- Answer: We completely agree with the reviewer and rephrased the passage to avoid ageism.
Reviewer: "Line 42-51: Reports evidence of the effects of music therapy in people diagnosed with AD... However, there is a history of previously executed work that should be updated.".
- Answer: We discussed the new studies in the text and updated the references.
Reviewer: "... it is suggested to include the descriptive statistics obtained from each group in order to have references to the cognitive state of the participants. ... Line 172-173. ... This clinical or experimental group should be described in very precise terms and only once, to avoid confusion. .... Line 176: What diagnostic condition did the older adults considered in the experimental group have?"
- Answer: In the participants section, the description of the clinical and the healthy group was reformulated to avoid confusion, and descriptive statistics for both groups were included.
Reviewer: "Line 177: It says that the participants completed 75% of the treatment sessions. It is advisable to reference the criterion or bibliographic basis that supports that said 75% was sufficient to impact the results."
- Answer: This information was misplaced. Thus, we changed the text and included this information in the correct context in the 'procedures section.' The criterion's rationale is more evident now that the information is appropriately contextualized.
Reviewer: "The number of scales and tests for adults with cognitive impairment is striking. What was the order of administration, is it according to Table 1? Could fatigue or cognitive load have played a role at some point? How did you control these aspects?"
- Answer: We included information in the instruments and procedures sections to clarify when and how the tests were administrated. It is important to notice that not all the tests were applied to older adults. The physician provided some; others were applied to the family caregivers. We included this information in Table 1. The tests for the older adults were applied on two sessions of about 40 minutes each on the pre-test and re-applied on another two sessions of about 40 minutes each on the post-test. The order of the tests was counterbalanced, as described in the procedures session.
Reviewer: "Line 330-335: The stimulated neuropsychological functions mentioned are 10. What relationship did the WM procedures have with the tests used to evaluate these functions, since Table 1 presents several, more than ten tests or scales?"
- Answer: Assuming that WM refers to Working Memory, there are two tests in Table 1 directly related to Working Memory: WAIS-III (digits subtest -> phonological working memory) and The Corsi Block-tapping test (visuospatial working memory).
Reviewer: "....on Line 337: One per day, or did it have another distribution? In relation to the test administration times, what was the delay in this regard?"
- Answer: This information was included in the procedures section. In both the pre-test and the post-test, all the instruments were applied in two sessions of about 40 minutes each. The post-test occurred about six weeks after the pre-test. The experimental group attended two weekly sessions of music therapy (with a maximum of one session per day), totaling 12 sessions of approximately 30 to 40 minutes each.
Reviewer: "The graphs should have the error bars that can be obtained from the application of the 3-way ANOVA in a statistical program."
- Answer: We added the error bars in the graphs as requested.
Reviewer: "What background allows us to discuss this result and this interpretation? Is it possible to compare it with evidence obtained in other studies?" and ... "The question is whether this TM favors those types of associations that allow it to respond or maintain functionality, which is key in terms of episodic memory."
- Answer: We altered different parts of the discussion to make clear the role of "musical encoding" on episodic memory. First, we compared our results with previous studies with converging results, like Palisson et al. [62], and Basaglia-Pappas et al. [63]. Then, we also discussed why this strategy (musical encoding) would not benefit autobiographical memory.
Reviewer: "Another question that remains, is whether there is evidence in the literature that in the 5 tests mentioned there are no significant differences compared to a powerful stimulation such as MT."
- Answer: One of the problems of the literature concerning music therapy's effects on memory is that most studies have not employed specific memory tests [see 2, 15, 16, 17, 23 for reviews of this issue]. Thus, there is no clear evidence in the literature about this issue. On the other hand, the number and the types of tests used in the present study are one of its strengths. We pointed out this issue in the introduction.
Reviewer: In the discussion and conclusion sections, the reviewer raises concerns about the role of WM (Working Memory) on episodic memory and the intervention effects. For example: "...Although the contents of the tests are not directly related to WM, could the training of cognitive processes through WM strategies favor them?" "...The question is ... what aspects of episodic memory could its role in WM be focused on that allow the results obtained in people with AD?" "...How does WM contribute to episodic memory, as observed?"
- Answer: The reviewer assumed that the music therapy intervention affected Working Memory and asked if this improvement in working memory could somehow mediate the improvement in episodic memory. However, the music therapy intervention did not affect working memory in the present study (as well as in other studies; see reference 23). Thus, it cannot be a causal explanation for the changes in episodic memory. We checked and made explicit in the results section that "...because of the design we used, the effect of the treatment (experimental vs. control) was assessed by specific tests only when the time X group interaction (or any higher order interaction including these factors) was statistically significant. In none of the working memory or autobiographical memory tests, there was a statistically significant time X group interaction or other higher order interaction including time X group.".
Reviewer: "was their life-historical relationship with music considered in the previous study of the participants? And with what kind of music? Or is it just some aspects such as rhythms, tones, and harmonies, among other aspects of music?"
- Answer: We addressed this question in the procedures section. We made clear in the text that the participants' history with music was assessed and considered in the music therapy intervention.
Reviewer: "Line 615-621: Was it considered a way to evaluate the role of the caregiver, mediator, or family member, in terms of their TM intersession intervention?"
- Answer: It was explained in the instruments and procedures sections that the caregivers were instructed to keep the musical stimulation at home in the last session of the music therapy interventions. In addition, the caregivers answered the Katz Functional Assessment Scale and the Bayer Scale. Thus, in the discussion section, we decided to restrict ourselves to the results of the intervention on the tests.
Reviewer: "Finally, it is suggested that the conclusion be oriented towards the findings of this research, mentioned punctually and clearly."
- Answer: We reformulated the conclusion, making it shorter and more focused.
Sincerely,
the authors
Reviewer 3 Report
Comments and Suggestions for Authors
Your article was insightful and a very interesting topic choice. There are a items that do need to be revised or further explored:
Line 44, can you expand on the meaning of "practicing music"
After reading further into the manuscript (line 339), it seems to be first noted that the music therapist who conducted the MT is also a psychologist. This would be better if introduced at the beginning.
At times, when the therapist who conducted the intervention is discussed it is noted that the therapist is the first author sometimes and sometimes not-this can come off as it being a different person based on how the therapist is mentioned.
Briefly discussing the background, qualifications, and training of the "trained research assistants" would provide better clarity on who is performing the pre-and post tests.
The term "older age" is noted throughout the manuscript. In the inclusion criteria it mentions "being over 60 years old" as well as "older adults who can be accompanied...." Does the term "older age" refer to a certain age group separate from those 60 years and older?
There were several tools utilized to measure the intervention that was implemented in the study. Providing validity and reliability of each tool would support the strength of the tool as some of the tools mentioned were developed by the author or may be unknown to the reader.
Many of the references used were more than fives years old. Using resources that were written within the last five years assist to exhibit the utilization of updated information in the search for literature relating to your topic.
Comments on the Quality of English Language
There are grammatical errors noted throughout the paper. For example, line 79 should read "indirectly" instead of "direct". Line 269 should read "them" and not "then".
There are a few sentences noted throughout the paper that are repetitive word for word. A few examples are lines 103-106 and lines 163-166; lines 274-277 and 340-344.
Author Response
Dear reviewer,
Thank you for your encouraging review. In the manuscript's revised version, we incorporated all your suggestions. We included:
- We made it clear the meaning of practicing music.
- We made it clear that the music therapy was conducted by the first author of the present study, a certified music therapist and a psychologist. This information was included in the procedures section.
- In the procedures section, we included more information on the background of the research assistants.
- We removed the term older age and standardized older adults' use throughout the manuscript.
- In the instruments section, we included references about the validity and reliability of the standardized measures used in the research. In addition, for the measures that we developed, we calculated and provided the test-retest reliability of these measures.
- We updated the references, including several studies from the last five years.
Sincerely,
the authors
Round 2
Reviewer 2 Report
Comments and Suggestions for Authors
Dear author
The revised aspects comply with what was suggested, greatly improving the study. Specifically, they adjust precision when referring to the adult, the tests used, the clinical data, the relevant graphs and statistics, and the aspects indicated in the discussion and conclusions. In summary, it is approved for publication.